# Mental health among healthcare workers during COVID-19 pandemic in Thailand

**Chotiman Chinvararak**[ID]**, Nitchawan Kerdcharoen, Wisarat Pruttithavorn**[ID]**, Nongnuch Polruamngern, Thanin Asawaroekwisoot, Wanida Munsukpol, Pantri Kirdchok**[ID]*

Department of Psychiatry, Faculty of Medicine Vajira Hospital, Navamindradhiraj University, Bangkok, Thailand

* pantri@nmu.ac.th

**Data Availability Statement:** All relevant data are within the paper and its Supporting Information files.

## Abstract

### Objectives

This study aimed to measure the prevalence of burnout syndrome, anxiety, depression, and post-traumatic disorders (PTSD), as well as examine their associated factors among Thai healthcare workers (HCWs) during COVID-19 outbreak.

### Method

We employed a multiple-method design at a tertiary-care hospital in Bangkok between May 22, 2021 and June 30, 2021 by using an online survey. The information included demographic characteristics, work details, perceived support, PTSD symptoms, Maslach Burnout Inventory: General Survey (MBI-GS), General Anxiety Disorder-7 (GAD-7), Patient Health Questionnaire (PHQ-2 and PHQ-9), and narrative response to an open-ended question. The associated factors of mental health problems were analysed by multiple logistic regression analyses. The qualitative data were analysed by the content analysis method.

### Results

A total of 986 HCWs (89.1% female; mean age = 34.89 ± 11.05 years) responded to the survey. 16.3%, 16%, and 53.5% of respondents had a high level of emotional exhaustion, depersonalisation, and diminished personal achievement, respectively. 33.1%, 13.8%, and 2.3% of respondents had anxiety, depression, and PTSD. Risk factors of emotional exhaustion were male sex ($OR_{adj}$ = 2.29), nurses ($OR_{adj}$ = 3.04), doctors ($OR_{adj}$ = 4.29), working at COVID-19 inpatient unit ($OR_{adj}$ = 2.97), and working at COVID-19 intensive care unit ($OR_{adj}$ = 3.00). Additionally, preexisting mental illness was associated with anxiety ($OR_{adj}$ = 2.89), depression ($OR_{adj}$ = 3.47), and PTSD ($OR_{adj}$ = 4.06). From qualitative analysis, participants reported that these factors would improve their mental health: supportive and respectful colleagues, appropriate financial compensation, reduced workload, clarity of policy and communication channel, and adequate personal protective equipment.

**Funding:** Navamindradhiraj University Research Fund.

**Competing interests:** The authors have declared that no competing interests exist.

## Conclusions

Thai HCWs experienced negative mental health outcomes during the COVID-19 pandemic substantially. This issue needs attention and actions should be implemented to support them.

## Introduction

The COVID-19 pandemic has been a challenge to all populations and public health systems around the globe. Since the World Health Organization declaration in March 2020 [1] the pandemic situation has been fluctuating, and the crisis has yet to be fully resolved.

It has impacted not only lifestyles, the economy, and physical health but also the mental health of individuals. High rates of psychological distress, stress, anxiety, depression, and post-traumatic stress disorder (PTSD) were reported in general populations in many countries in middle-income countries and higher-income countries on different continents [2–4]. However, prompt government implementation of stringent measures reduced the depression rate [5]. Among the middle-income countries, Thailand had the highest degrees of adverse mental health in the general population [6]. Physical symptoms suggesting COVID-19 infection were positively associated with high mental health outcomes [3]. Vice versa, having an underlying mood disorder increases the risk of COVID-19 hospitalisation and mortality [6].

In Thailand, since its first documented COVID-19 case in January 2020, the situation has continued to worsen even though Thailand was considered to be efficient in containing the COVID-19 in its early stage of spread [7]. To demonstrate, since the third wave of the outbreak in Thailand in April 2021, the number of accumulative COVID-19 cases has been increasing continuously. On April 30, 2021, there were 62,153 confirmed COVID-19 cases, and by the end of June 2021, the number rose to 259,301, most of which were in the Bangkok Metropolitan Region [8]. Healthcare systems are also varying between countries. During this study period, Thailand has not implemented a home isolation protocol, and all patients with COVID-19 were admitted to hospitals or 'Hospitels', a government-run medicalised hotels for COVID-19 patients.

The pandemic impacts the work routines, work-related stress, and personal life of healthcare workers (HCWs) [9]. They are at higher risk of exposure to COVID-19 infection [10], in which doctors reported as more likely to exposure than nurses [11]. Extended working hours during the pandemic could lead to burnout and other adverse psychological problems. The psychological outcomes of HCWs during the pandemic have been studied globally. Common problems consisted of burnout syndrome, anxiety, depression, and PTSD [12–23]. Pooled prevalences of anxiety, depression, acute stress, post-traumatic stress are 30.0, 31.3, 56.5, and 20.2% respectively [18]. Studies in Asia showed the varied proportions of these problems in each country; anxiety 8.7–90%; depression 0.8–58%; post-traumatic stress 2.1–9.1% [19–22]. The pooled prevalences of anxiety and depression among East Asia countries are 20.5 and 19.1%, respectively [23]. With these psychological health impacts, not only will healthcare 'workers' well-being be undermined, but their decision-making skills that involve patients' care will be compromised. These unpleasant conditions will eventually affect not only patients but also society as a whole.

A number of studies also identified associated factors of negative mental health outcomes among HCWs. Regarding burnout, studies identified risk factors of burnout, including younger age, high workload [13], occupational factors, gender differences, [14] working on

frontlines [15], shortage of resources [16]. Regarding associated factors of anxiety, depression, and PTSD among HCWs, results are inconclusive. However, potential risk factors include younger age, female, working frontline, lack of personal protective equipment, nurse or doctors, knowing someone who died of COVID-19, and lack of support were found to be associated in several studies [24–29]. Studies in Asia also revealed non-medical HCWs, physical symptoms, pre-existing medical conditions, and old age are more likely to experience negative mental health outcomes [10, 19–21]. To our knowledge, even though there was a study in Thailand found that 42.5% of healthcare workers had anxiety [30], there has yet to be a study on burnout syndrome, depression, and post-traumatic disorders among healthcare workers in Thailand. Furthermore, the COVID-19 pandemic situation changes continuously, and healthcare systems are specific to their countries. Therefore, we aim to measure the prevalence of burnout syndrome, anxiety, depression, and post-traumatic disorders, as well as their potential risks and protective factors among Thai healthcare workers during the ongoing third wave of COVID-19 outbreak in Thailand. Furthermore, the qualitative part aims to explore healthcare workers' perspectives on factors that can improve their mental well-being.

## Method

### Study design, setting and participants

This study, conducted between May 22, 2021 and June 30, 2021, employs a multiple-method research design by using an online survey to gather quantitative data and narrative responses from an open-ended question. Inclusion criteria were healthcare workers working in any position at a tertiary-care hospital in Bangkok, which has been the center of the pandemic in Thailand. Hospital services relevant to COVID-19 treatment were medical evaluation for people at risk, inpatient units, intensive care units, and medical services at a Hospitel a new medical term that refers to a medicalised hotel for COVID-19 patients to receive treatment and recover.

Regarding participants, the proportions of healthcare workers with burnout syndrome, anxiety, and depression during the COVID-19 pandemic were estimated to be 0.34 [13], 0.41 [12], and 0.31 [12]. With error (d) = 0.05 and a finite population of 3218, the required sample size was 334 [31]. Research participants were approached through LINE application, the main social media platform connecting all healthcare workers of the hospital together. The convenience-based sampling method was used and the data were collected using Google Forms software. All participants were informed of the study's objectives, method, and a consent statement before starting the survey. The ethical approval for the study protocol was officially granted by the Ethical Review Committee of the University (COA134/2564).

### Measurements

The study instruments consisted of a questionnaire comprising demographic characteristics, work details, perceived adequate support from hospital and colleagues, and financial compensation. The questionnaire about perceived support used 5-point Likert scale, with 1 as strongly disagree and 5 as strongly agree.

Burnout syndrome was measured by the Thai version of the Maslach Burnout Inventory: General Survey (MBI-GS) which contains 16 items with three dimensions: emotional exhaustion (EE, 5 items; Cronbach's alpha coefficient = 0.9), depersonalisation (DP, 5 items; Cronbach's alpha coefficient = 0.7), and diminished personal accomplishment (PA, 6 items; Cronbach's alpha coefficient = 0.7) [32, 33]. Subscales scores were considered as low, moderate or high level of burnout syndrome according to these cut-points; low EE $\leq 10$, DP $\leq 5$, PA $\geq 30$; moderate EE = 11–15, DP = 6–10, PA = 25–29; high EE $\geq 16$, DP $\geq 11$, PA $\leq 24$ [32].

Anxiety was measured by the Thai version of General Anxiety Disorder-7 (GAD-7) questionnaire. The scores were interpreted as followed: normal (0–4); mild (5–9); moderate (10–14); and severe (15–21). The cut-point for having anxiety was five yielding sensitivity and specificity, 89% and 82% respectively to detect generalised anxiety disorder [34].

Depression was measured with 2-step approach using the Thai Patient Health Questionnaire (PHQ-2 and Thai PHQ-9). Respondents who score ≥ 1 from PHQ-2 will be asked to answer PHQ-9. The scores were interpreted as follows: normal (0–6), mild (7–12), moderate (13–18), and severe (≥19). As Thai PHQ-9 score ≥ 7 is 95% sensitive and 55% specific for a diagnosis of major depression, the cut-point for having depression was seven [35, 36].

Post-traumatic stress disorder (PTSD) was measured by 3-questions, which we adapted from DSM-5 criteria [37]; presence of work-related trauma, presence of avoidance of relevant stimuli/hypervigilance or re-experiencing symptoms, and impaired function. All criteria must be met in order for participants to be identified as having PTSD.

At the end of the questionnaire, participants were addressed the confidentiality of the survey and asked the open-ended question; What factors will improve your mental health or reduce your stress? The participant's narrative answer to this question will be analysed with qualitative methods.

## Data analyses

Quantitative data analysis was performed using Stata version 14.0 (StataCorp, College Station, TX, USA). Descriptive statistics were used to explore 'HCWs' characteristics and their mental health outcomes. The associations between the outcomes (burnout syndrome, anxiety, depression, and PTSD) and variables were assessed by the Chi-square test, ' 'Fisher's exact test, independent sample t-tests, or Mann–Whitney U tests. Binary logistic regression, followed by multiple logistic regression were used to calculate the odds ratios. Variables were included in the multivariable model if they have a p-value < 0.05 in univariable analysis.

As for qualitative data, answers to open-ended questions were analysed using content analysis. Two trained researchers independently coded the responses by means of inductive analysis, using the occurrence of themes and subthemes from texts. If inconsistency occurs, the researchers compared the analyses to reach a consensus before extracting statements that best represented each identified theme and subthemes.

## Results

Of 3,218 hospital HCWs, 986 (31.5%) responded to the survey. The respondents included 623 nurses, 50 doctors, 173 allied health professionals, and 140 support staff. The majority of respondents across the positions mentioned above were female, and the mean age was 34.89 ±11.05. The median working hours per week (IQR) was 48 (40–56) hours. Interestingly, only 2.5% of respondents wished to receive counseling from the psychologist/psychiatrist. Other work characteristics and perceived support are presented in Table 1.

### Prevalence of burnout and mental problems

Regarding the prevalence of burnout, 16.3%,16% and 53.5% of respondents had high level of EE, DP, and diminished PA, respectively. However, the figures of nurses, doctors and HCWs at COVID-19 inpatient units were higher as followed: Nurse (EE = 19.7%, DP = 17.8%), doctors (EE = 38.0%, DP = 44.0%), non-intensive care unit (EE = 27.5%, DP = 23.1%), intensive care unit (EE = 40.2%, DP = 31.7%). Additionally, overall, 16.2% had high level of burnout in 2 or more domains (Table 2).

**Table 1. Demographic, work characteristics, and perceived support (n = 986).**

| | N (%) | |
|---|---|---|
| Sex | | |
| Male | 107 | (10.9) |
| Female | 879 | (89.1) |
| Age, year, Mean (SD) | 34.89 (11.05) | |
| Occupation | | |
| Nurse | 623 | (63.2) |
| Doctor | 50 | (5.1) |
| Allied health professional | 173 | (17.5) |
| Support staff; administrative, technicians, security, cleaners | 140 | (14.2) |
| Have a chronic medical condition | 255 | (25.9) |
| Have mental health illness | 41 | (4.2) |
| Current working unit | | |
| Non-Covid-19 related | 358 | (36.3) |
| OPD for patients with high-risk but unconfirmed COVID-19, parttime | 252 | (25.6) |
| OPD for patients with high-risk but unconfirmed COVID-19, fulltime | 82 | (8.3) |
| Non-ICU, with confirmed COVID-19 patients | 91 | (9.2) |
| ICU, with confirmed COVID-19 patients | 82 | (8.3) |
| Hospitel, with confirmed COVID-19 patients | 121 | (12.3) |
| Working hour (hours/week) median (IQR) | 48 | (40–56) |
| Have been transferred from other units (yes) | 183 | (18.6) |
| Intended not to go home (yes) | 705 | (71.5) |
| Perceived adequate support from hospital | 3.51 ± 0.95 | |
| Perceived adequate support from collogues | 4.04 ± 0.84 | |
| Perceived adequate financial compensation | 2.70 ± 1.21 | |
| Wish to receive a counseling from a psychologist/psychiatrist | | |
| Yes | 25 | (2.5) |
| No | 948 | (96.1) |
| Uncertain | 13 | (1.3) |

Data are presented as number (%), mean ± standard deviation or median (interquartile range).

Abbreviations: OPD, outpatient department; IPD, inpatient department; ICU, intensive care unit.

With respect to anxiety, depression, and post-traumatic stress disorder, 33.4% of HCWs had anxiety, comprising 27.1%, 3.4%, 0.9% of mild, moderate and severe anxiety consecutively. 13.8% of HCWs had depression, comprising 10.7%, 2.1%, 1% of mild, moderate, severe depression consecutively. Finally, 2.2% of respondents were considered to have PTSD. Like burnout, these figures of nurses, doctors and HCWs working at COVID-19 inpatient units were higher (Table 2).

## Associated factors of burnout

From univariate analysis, we found that sex and perceived support were associated with all domains of burnout. Age, occupation, working unit, working hour, and being transferred were associated with EE. Occupation, mental illness, working unit, working hour were associated with DP. Occupation, working unit were associated with PA.

From multivariable analysis, the male sex was associated with DP and PA ($OR_{adj}$ = 2.29 and 1.69). Nurses and doctors were risk factors of EE ($OR_{adj}$ = 3.04 and 4.29) and DP ($OR_{adj}$ = 2.74 and 4.61) Working at COVID-19 inpatient unit increased the risks of EE (non-intensive care

**Table 2. Prevalence of burnout and mental health problems.**

| Variables | N (%) | | | | |
|---|---|---|---|---|---|
| | All HCWs (n = 986) | | Nurses (n = 623) | Doctors (n = 50) | IPD COVID (n = 91) | ICU COVID (n = 82) |
| Burnout syndrome | | | | | |
| High EE | 161 | (16.3) | 123 (19.7) | 19 (38.0) | 25 (27.5) | 33 (40.2) |
| High DP | 158 | (16.0) | 111 (17.8) | 22 (44.0) | 21 (23.1) | 26 (31.7) |
| High PA | 527 | (53.5) | 300 (48.2) | 26 (52.0) | 47 (51.6) | 48 (58.5) |
| ≥ 2 types of high burnout | 160 | (16.3) | 117 (18.8) | 11 (40.0) | 24 (26.4) | 34 (41.5) |
| Anxiety disorder (GAD-7) | | | | | |
| Yes (≥5) | 309 | (31.3) | 196 (31.5) | 22 (44.0) | 25 (27.5) | 42 (51.1) |
| Depressive disorder | | | | | |
| PHQ-9 (n = 312) | | | | | |
| Yes (≥7) | 136 | (13.8) | 87 (14.0) | 16 (32.0) | 11 (12.1) | 18 (22.0) |
| Post-traumatic stress disorder | | | | | |
| Yes | 22 | (2.2) | 11 (1.8) | 5 (10.0) | 4 (4.4) | 3 (3.66) |

Abbreviations: EE, emotional exhaustion; DP, depersonalisation; PA, personal accomplishment; HCWs, healthcare workers; IPD, inpatient unit; ICU, intensive care unit.

unit; $OR_{adj}$ = 2.97, intensive care unit; $OR_{adj}$ = 3.00), while working at Hospitel reduced the risk ($OR_{adj}$ = 0.68). Mental illness and working hour were associated with DP. Perceived support from hospital reduced risks of EE and DP and perceived support from colleagues reduced the risk of PA (Table 3).

## Associated factors of mental problems

From univariate analysis, male sex and perceived support were associated with all mental health problems. Mental illness, working unit, working hour, and being transferred were associated with anxiety. Occupation, medical condition, mental illness, working unit, working hour were associated with depression. Medical conditions, mental illness, and working hour were associated with PTSD.

From multivariable analysis, male sex was associated with anxiety and PTSD ($OR_{adj}$ = 1.6 and,4.05). Having mental illness was associated with anxiety ($OR_{adj}$ = 2.89), depression ($OR_{adj}$ = 3.47), and PTSD ($OR_{adj}$ = 4.06). Working at the COVID-19 intensive care unit increased the risk of anxiety ($OR_{adj}$ = 2.07). Having 49–56 working hours/week increased the risk of PTSD ($OR_{adj}$ = 6.93). Perceived adequate hospital support reduced the risk of anxiety and depression. Occupation and chronic medical conditions were not associated with any mental problems (Table 4).

## Qualitative results

Of 986 respondents, 221 (25%) provided narrative responses to 'What factors will improve your mental health or reduce your stress?'. The answers related to work can be divided into 5 areas; 1) colleagues 2) financial compensation 3) workload 4) organisation management and policy 5) personal protective equipment (PPE).

**1) Colleagues.** 43 out of 211 respondents (19.4%) said colleagues were a key factor in mental health conditions. The two main kinds of colleagues most desired by respondents were colleagues who can help each other and colleagues who are respectful and nonjudgmental.

**Table 3. Adjusted odds ratios from multivariable analysis of the associated factors of burnout among HCWs.**

| Factors | ≥ 2 domains of high burnout | | | Emotional exhaustion | | | Depersonalisation | | | Personal achievement | | |
|---|---|---|---|---|---|---|---|---|---|---|---|---|
| | OR$_{adj}$ | 95%CI | p-value | OR$_{adj}$ | 95%CI | p-value | OR$_{adj}$ | 95%CI | p-value | OR$_{adj}$ | 95%CI | p-value |
| Sex | | | | | | | | | | | | |
| Male | 2.10 | (1.11–4.00) | 0.023* | 1.91 | (1.00–3.66) | 0.051 | 2.29 | (1.25–4.20) | 0.008* | 1.69 | (1.05–2.73) | 0.032* |
| Female | 1.00 | Reference | | 1.00 | Reference | | 1.00 | Reference | | 1.00 | Reference | |
| Age, year, Mean (SD) | 0.98 | (0.96–1.00) | 0.026* | 0.97 | (0.95–0.99) | 0.001* | | | | | | |
| Occupation | | | | | | | | | | | | |
| Nurse | 2.95 | (1.32–6.56) | 0.008* | 3.04 | (1.35–6.84) | 0.007* | 2.74 | (1.34–5.62) | 0.006* | 0.70 | (0.47–1.05) | 0.086 |
| Doctor | 5.05 | (1.94–13.1) | 0.001* | 4.29 | (1.62–11.39) | 0.003* | 4.61 | (1.91–11.11) | 0.001* | 0.56 | (0.28–1.12) | 0.099 |
| Allied health professional | 0.87 | (0.33–2.31) | 0.780 | 0.81 | (0.30–2.20) | 0.678 | 1.10 | (0.45–2.64) | 0.840 | 1.53 | (0.94–2.48) | 0.087 |
| Support staff | 1.00 | Reference | | 1.00 | Reference | | 1.00 | Reference | | 1.00 | Reference | |
| Mental illness | | | | | | | | | | | | |
| No | | | | | | | 1.00 | Reference | | | | |
| Yes | | | | | | | 2.70 | (1.28–5.70) | 0.009* | | | |
| Current working unit | | | | | | | | | | | | |
| Non-Covid-19 | 1.00 | Reference | | 1.00 | Reference | | 1.00 | Reference | | 1.00 | Reference | |
| Non-ICU COVID-19 IPD | 2.79 | (1.39–5.58) | 0.004* | 2.97 | (1.50–5.87) | 0.002* | 1.71 | (0.96–3.05) | 0.071 | 0.88 | (0.56–1.38) | 0.576 |
| ICU COVID-19 IPD | 3.01 | (1.60–5.63) | 0.001* | 3.00 | (1.61–5.59) | 0.001* | 1.48 | (0.84–2.62) | 0.177 | 1.31 | (0.80–2.12) | 0.282 |
| Hospitel | 0.51 | (0.17–1.5) | 0.220 | 0.49 | (0.17–1.43) | 0.190 | 0.34 | (0.13–0.88) | 0.027* | 1.32 | (0.87–2.01) | 0.185 |
| Working hour (hours/week) | | | | | | | | | | | | |
| ≤40 | 1.00 | Reference | | 1.00 | Reference | | 1.00 | Reference | | | | |
| 41–48 | 0.91 | (0.52–1.59) | 0.733 | 1.08 | (0.62–1.88) | 0.786 | 1.02 | (0.59–1.76) | 0.954 | | | |
| 49–56 | 1.28 | (0.75–2.18) | 0.359 | 1.53 | (0.91–2.59) | 0.113 | 1.66 | (0.99–2.77) | 0.049* | | | |
| >56 | 0.91 | (0.56–1.49) | 0.707 | 1.21 | (0.75–1.96) | 0.437 | 1.17 | (0.73–1.88) | 0.513 | | | |
| Transfer from other units | | | | | | | | | | | | |
| No | 1.00 | Reference | | 1.00 | Reference | | | | | | | |
| Yes | 0.76 | (0.46–1.27) | 0.294 | 0.73 | (0.44–1.21) | 0.218 | | | | | | |
| Support from hospital | 0.56 | (0.43–0.73) | <0.001* | 0.68 | (0.53–0.87) | 0.003* | 0.65 | (0.51–0.83) | 0.001* | 0.86 | (0.73–1.02) | 0.087 |
| Support from collogues | 1.02 | (0.80–1.31) | 0.858 | 1.09 | (0.85–1.39) | 0.496 | 1.01 | (0.80–1.29) | 0.914 | 0.82 | (0.68–0.99) | 0.037* |
| Financial compensation | 0.90 | (0.74–1.1) | 0.298 | 0.86 | (0.71–1.04) | 0.116 | 0.92 | (0.76–1.11) | 0.387 | | | |

Abbreviations: ORadj, Adjusted Odds Ratio; CI, confident interval. Variable was included in multivariable model due to have p-value < 0.050 in univariable analysis.

"Having colleagues work together harmoniously, using humble words, honoring and respecting for each other and sharing responsibility. Good colleagues improve our mental health" (Nurse, 30 years old).

"I am lucky to have a close friend at work who can discuss and help each other in everything" (Nurse, 47 years old).

**2) Financial compensation.** Financial compensation was reported by 35 respondents (15.8%).

"It would be better if there was a compensation worth the risk according to the workload received. It is discouraged that the pay is not worth the hard work" (Nurse, 40 years old)

"Reasonable compensation! Now, it feels like working hard without being paid. Everyone should be paid for their hard work. Now I feel that I'm being taken advantage" (Doctor, 28 years old)

**Table 4. Adjusted odds ratios from multivariable analysis of the associated factors of anxiety, depression, and PTSD among HCWs.**

| Factors | Anxiety | | | Depression | | | PTSD | | |
|---|---|---|---|---|---|---|---|---|---|
| | OR$_{adj}$ | 95%CI | p-value | OR$_{adj}$ | 95%CI | p-value | OR$_{adj}$ | 95%CI | p-value |
| Sex | | | | | | | | | |
| Male | 1.65 | (1.07–2.56) | 0.025* | 1.67 | (0.90–3.10) | 0.104 | 4.05 | (1.10–14.95) | 0.036* |
| Female | 1.00 | Reference | | 1.00 | Reference | | 1.00 | Reference | |
| Occupation | | | | | | | | | |
| Nurse | | | | 1.20 | (0.64–2.26) | 0.573 | 0.85 | (0.19–3.71) | 0.825 |
| Doctor | | | | 1.92 | (0.81–4.53) | 0.138 | 1.64 | (0.33–8.11) | 0.545 |
| Allied health professional | | | | 0.79 | (0.36–1.74) | 0.557 | 0.21 | (0.02–2.09) | 0.182 |
| Support staff | | | | 1.00 | Reference | | 1.00 | Reference | |
| Chronic medical condition | | | | | | | | | |
| No | | | | 1.00 | Reference | | 1.00 | Reference | |
| Yes | | | | 1.41 | (0.92–2.15) | 0.112 | 1.78 | (0.66–4.80) | 0.254 |
| Mental illness | | | | | | | | | |
| No | 1.00 | Reference | | 1.00 | Reference | | 1.00 | Reference | |
| Yes | 2.89 | (1.50–5.57) | 0.002* | 3.47 | (1.69–7.12) | 0.001* | 4.06 | (1.09–15.12) | 0.037* |
| Current working unit | | | | | | | | | |
| Non-Covid-19 | 1.00 | Reference | | 1.00 | Reference | | | | |
| Non-ICU COVID-19 IPD | 0.90 | (0.50–1.63) | 0.735 | 0.91 | (0.45–1.84) | 0.783 | | | |
| ICU COVID-19 IPD | 2.07 | (1.19–3.58) | 0.010* | 1.10 | (0.58–2.09) | 0.768 | | | |
| Hospitel | 1.18 | (0.64–2.17) | 0.590 | 0.65 | (0.31–1.37) | 0.262 | | | |
| Working hour (hours/week) | | | | | | | | | |
| ≤40 | 1.00 | Reference | | 1.00 | Reference | | 1.00 | Reference | |
| 41–48 | 1.41 | (0.95–2.10) | 0.089 | 1.53 | (0.88–2.65) | 0.135 | 4.30 | (0.89–20.94) | 0.071 |
| 49–56 | 1.25 | (0.83–1.89) | 0.282 | 1.43 | (0.81–2.52) | 0.212 | 6.93 | (1.58–30.4) | 0.010* |
| >56 | 0.93 | (0.64–1.36) | 0.711 | 1.58 | (0.97–2.57) | 0.067 | 3.58 | (0.86–14.93) | 0.080 |
| Transfer from other units | | | | | | | | | |
| No | 1.00 | Reference | | | | | | | |
| Yes | 1.29 | (0.88–1.91) | 0.194 | | | | | | |
| Support from hospital | 0.72 | (0.59–0.87) | 0.001* | 0.67 | (0.52–0.86) | 0.002* | 0.56 | (0.29–1.08) | 0.085 |
| Support from collogues | 0.88 | (0.73–1.07) | 0.206 | 1.06 | (0.83–1.36) | 0.636 | 0.58 | (0.34–1.00) | 0.051 |
| Financial compensation | 0.99 | (0.86–1.14) | 0.866 | 0.81 | (0.67–0.98) | 0.034 | 1.02 | (0.58–1.81) | 0.940 |

Abbreviations: ORadj, Adjusted Odds Ratio; CI, confident interval. Variable was included in multivariable model due to have p-value < 0.050 in univariable analysis.

**3) Workload.** This issue was reported by 38 respondents (17.2%). The respondents wished for reduced workload, an increase in manpower or a circulation of healthcare workers to take care of COVID-19 patients, and more free time.

"Please reduce unrelated workload, for example, less paperwork" (Nurse, 43 years old)

"We need additional manpower. Hired more workers. Now, the patients are increasing while manpower remains unchanged" (Nurse, 49 years old)

"Having time to talk to someone could help ease the pressure and stress. I am still unsure if I should seek advice because I'm stressed but I am so busy working that I don't think I have time for counseling" (Nurse, 48 years old)

**4) Management and policy.** This was reported by 27 respondents (12.0%). Management and policy-related issues included clarity in policy and communication and feedback pathway.

"Work or relocation should be notified early to allow time for preparation. The nature of the work descriptive should be clear" (Nurse, 46 years old)

"I wish executives meet and talk to encourage (workers) at the worksite more" (Doctor, 55 years old)

"I wish the executives visit the frontline personally. A policy, that empathies workers, notified clearly" (Doctor, 28 years old)

**5) Personal Protective Equipment (PPE).** The issue was reported by 26 respondents (11.8%) who believed that having enough personal protective equipment would improve their mental health.

"Protective devices should be available fully, such as N95, hair cap, etc. Now they are scarce and insufficient, causing anxiety and insecurity while taking care of high-risk patients" (Nurse, 37 years old)

"Healthcare workers should be protected more. PPE should be sufficient. Workers should not be left to seek them themselves. With sufficient equipment and personnel, the anxiety can be reduced" (Nurse, 28 years old)

## Discussion

To the best of our knowledge, this study is the first one to investigate the prevalence of burnout syndrome, depression, and post-traumatic stress disorder (PTSD) among Thai healthcare workers. From almost one thousand HCWs, the overall prevalence of burnout and PTSD was lower than that in the majority of studies but comparable to a study in India [12–15, 20]. The prevalence of anxiety and depression in our study is higher than in studies from Asia, which used the DASS-21 questionnaire as a measurement [10, 19–21]. However, the figures are smaller when compared to studies that used the GAD-7 and PHQ-9 questionnaires as we did [12, 27]. The smaller prevalence is suspected to result from the diversity of samples from various occupations and settings we included. Among those working at the intensive care unit, the prevalence of burnout was comparable to previous studies [16]. Also, emotional exhaustion and depersonalisation among nurses and doctors were also comparable to several studies [13]. However, it is noticeable that diminished personal accomplishment was high in all groups of research participants. This is likely because our study was conducted when the pandemic situation in Thailand began to be seriously exacerbated, leading to changes in the work routines of healthcare workers. As a consequence, Thai healthcare workers felt unprepared and incompetent.

Regarding risk factors of burnout, occupation and working units were the strongest associated factors of burnout. In line with previous studies, nurses, doctors, and those working with COVID-19 patients were at very high risk [13, 14]. From our study, those working at COVID-19 inpatients units both non-intensive care and intensive care unit (ICU) had approximately a threefold increase in risks of having emotional exhaustion. However, the risks did not increase for those working at a Hospitel. We did anticipate this result because patients admitted to a Hospitel were either asymptomatic or having mild severity of COVID-19. As a result, healthcare workers at a Hospitel would have lesser workload and exposure to patients.

Regarding risk factors for other mental health outcomes, we found from the multivariable analysis that the prominent risk factor was pre-existing mental illness. According to our study, underlying mental illnesses increased the risks of anxiety, depression, and PTSD, with $OR_{adj}$ 2.89, 3.47, and 4.06, respectively. This was congruence with studies from China [38, 39] and Italy [40, 41], and a multinational study from Asia [20]; however, the result is incongruent with a study from Lebanon [42]. As in previous studies, working in ICU increased the risk of anxiety [43, 44]. Nevertheless, we did not find a statistically significant association between occupation and anxiety, depression or PTSD [17]. We speculated that this was because mental illness and working unit played a more prominent role in our mixed-setting population.

Regarding gender and mental health, several studies highlighted gender differences in the psychological outcomes of HCWs [14, 45]. These findings had mixed results, but our result was consistent with studies from China [46, 47] that men had a greater chance of depersonalisation, lack of personal achievement, anxiety and PTSD. As the proportion of genders of HCWs in all occupations and settings were fairly similar in our study, we believe that male gender was an isolated risk factor in our sample.

Our study has emphasised the importance of support from hospital and colleagues, the issue of which was also addressed by previous studies [48, 49]. Perceived support from the hospital was negatively correlated with emotional exhaustion, depersonalisation, anxiety and depression, whereas perceived support from colleagues was negatively correlated with diminished personal accomplishment. Our qualitative analysis also pointed out modifiable associated factors; supportive and respectful colleagues, appropriate financial compensation, reduced workload, clarity of policy and communication channel, and adequate PPE. The findings suggest that organisational infectious preventive measures and PPE may reduce adverse psychological outcomes, which correlate with previous studies [50, 51]. Concerning financial support, while a study from Ghana found that tax-free salary reduced negative psychological impacts [52], direct studies on financial compensation's impact on mental health are still lacking. Considering that the small proportion of healthcare workers did request counseling despite the high prevalence of mental problems in our study, actionable measures should be taken into account to prevent and reduce mental health problems among healthcare workers. However, among psychological treatments, cognitive behavioural therapy (CBT) is the most evidence-based treatment against psychiatric symptoms [53]. Additionally, internet-based CBT could be more proper than face-to-face CBT since it could prevent the spread of infection during the COVID-19 pandemic [54, 55].

Additionally, the result from the qualitative data revealed some healthcare workers believed that having supportive colleagues and effective hospital policy, including providing appropriate financial compensation, sufficient PPE, and empathy from executives, could benefit them to relieve psychological stress. These findings also support the quantitative results.

We acknowledge several limitations of this study. Firstly, due to the ' 'study's descriptive design at a single point of time, we cannot draw a conclusion about causal relationships and longitudinal outcomes. Secondly, according to convenience-based sampling, selective bias might occur, as HCWs with mental health problems might not want to participate in the study because they did not have enough time or energy to do so. This could result in an underestimation of problems. Thirdly, as research participants answered our open-ended question by typing a response instead of being interviewed, the information gathered could possibly be not in-depth. Fourthly, we did not gather some factors that potentially are confounders such as knowing someone who die of COVID-19. Also, some particular specialist doctors such as surgeons may experience more psychological problems than others [29]. Lastly, due to the diversity of medical services, our results could only be generalised to psychological outcomes of Thai HCWs in a tertiary-care hospital.

In conclusion, Thai HCWs also experienced negative mental health outcomes substantially. Prominent risk factors of burnout included nurse, doctors, and working at COVID-19 inpatient unit. Significant risk factors of anxiety, depression, and PTSD was pre-existing mental illness. HCWs in our study proposed measures to remediate their stress. These issues of mental health problems should be examined further, and some practical solutions to these problems should be put into action promptly for the improvement of Thai HCWs mental health.

## Supporting information

**S1 Data.**
(DTA)

## Acknowledgments

The authors acknowledge the help of all the study participants and would like to give appreciation to Anucha Kamson for his assistance in statistical analysis.

## Author Contributions

**Conceptualization:** Chotiman Chinvararak, Nitchawan Kerdcharoen, Wisarat Pruttithavorn, Pantri Kirdchok.

**Formal analysis:** Chotiman Chinvararak, Pantri Kirdchok.

**Funding acquisition:** Nitchawan Kerdcharoen, Pantri Kirdchok.

**Investigation:** Chotiman Chinvararak, Nongnuch Polruamngern, Thanin Asawaroekwisoot, Wanida Munsukpol, Pantri Kirdchok.

**Methodology:** Chotiman Chinvararak, Nitchawan Kerdcharoen, Wisarat Pruttithavorn, Pantri Kirdchok.

**Project administration:** Wanida Munsukpol, Pantri Kirdchok.

**Validation:** Chotiman Chinvararak, Nitchawan Kerdcharoen, Wisarat Pruttithavorn, Pantri Kirdchok.

**Writing – original draft:** Chotiman Chinvararak, Pantri Kirdchok.

**Writing – review & editing:** Chotiman Chinvararak, Nitchawan Kerdcharoen, Wisarat Pruttithavorn, Nongnuch Polruamngern, Pantri Kirdchok.

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
