## [Decision Letter · Decision Letter 0]

31 Jan 2022

PONE-D-21-33387Mental health among healthcare workers during COVID-19 pandemic in Thailand: A mixed-method analysisPLOS ONE

Dear Dr. Pantri Kirdchok,

Thank you for submitting your manuscript to PLOS ONE. After careful consideration, we feel that it has merit but does not fully meet PLOS ONE’s publication criteria as it currently stands. Therefore, we invite you to submit a revised version of the manuscript that addresses the points raised during the review process.

We look forward to receiving your revised manuscript.

Kind regards,

Alison Wang

Academic Editor

PLOS ONE

Journal Requirements:

Reviewers' comments:

Reviewer's Responses to Questions

**Comments to the Author**

3. Have the authors made all data underlying the findings in their manuscript fully available?

Reviewer #1: Yes

Reviewer #2: Yes

4. Is the manuscript presented in an intelligible fashion and written in standard English?

Reviewer #1: Yes

Reviewer #2: Yes

5. Review Comments to the Author

**Reviewer #1:** I have the following comments for the authors to address. I am happy to review this paper again.

1) Under the Introduction, the authors stated " Since the World Health Organization declaration on March 2020 [1]

the pandemic situation has been fluctuating, and the crisis has yet to be fully resolved". It is important to talk about the impact of pandemic on mental health based on the following landmark studies (but not limited to):

A systematic review of COVID-19 on mental health

Impact of COVID-19 pandemic on mental health in the general population: A systematic review [published online ahead of print, 2020 Aug 8]. J Affect Disord. 2020;277:55-64. doi:10.1016/j.jad.2020.08.001

The impact of COVID-19 on three continents:

A chain mediation model on COVID-19 symptoms and mental health outcomes in Americans, Asians and Europeans. Sci Rep 11, 6481 (2021). https://doi.org/10.1038/s41598-021-85943-7

The impact of COVID-19 on developing countries:

The impact of COVID-19 pandemic on physical and mental health of Asians: A study of seven middle-income countries in Asia. PLoS One. 2021 Feb 11;16(2):e0246824. doi: 10.1371/journal.pone.0246824. PMID: 33571297.

Government response during the pandemic:

Government response moderates the mental health impact of COVID-19: A systematic review and meta-analysis of depression outcomes across countries. J Affect Disord. 2021 May 27;290:364-377. doi: 10.1016/j.jad.2021.04.050. Epub ahead of print. PMID: 34052584.

Worst outcome of COVID infection due to depression

Association Between Mood Disorders and Risk of COVID-19 Infection, Hospitalization, and Death: A Systematic Review and Meta-analysis. JAMA Psychiatry. 2021 Jul 28. doi: 10.1001/jamapsychiatry.2021.1818. Epub ahead of print. PMID: 34319365.

2) Under the paragraph, "A number of studies also identified associated factors of negative mental health outcomes

63 among HCWs". Since Thailand belongs the Asia-Pacific, please comment on the research performed in this region based on the following landmark studies but not limited to the following studies Under the discussion, the authors should compare their findings with the findings from the following Asian countries:

Perception Toward Exposure Risk of COVID-19 Among Health Workers in Vietnam: Status and Correlated Factors. Front Public Health. 2021 May 25;9:589317. doi: 10.3389/fpubh.2021.589317. PMID: 34113595; PMCID: PMC8185209.

The impact of COVID-19 pandemic on global mental health: From the general public to healthcare workers. Ann Acad Med Singap. 2021 Mar;50(3):198-199. doi: 10.47102/annals-acadmedsg.202189. PMID: 33855314.

Asian-Pacific perspective on the psychological well-being of healthcare workers during the evolution of the COVID-19 pandemic. BJPsych Open. 2020;6(6):e116. Published 2020 Oct 8. doi:10.1192/bjo.2020.98

A multinational, multicentre study on the psychological outcomes and associated physical symptoms amongst healthcare workers during COVID-19 outbreak [published online ahead of print, 2020 Apr 21]. Brain Behav Immun. 2020;S0889-1591(20)30523-7. doi:10.1016/j.bbi.2020.04.049

Psychological Impact of the COVID-19 Pandemic on Health Care Workers in Singapore [published online ahead of print, 2020 Apr 6]. Ann Intern Med. 2020;M20-1083. doi:10.7326/M20-1083

Impacts of COVID-19 on the Life and Work of Healthcare Workers During the Nationwide Partial Lockdown in Vietnam. Front Psychol. 2021 Aug 19;12:563193. doi: 10.3389/fpsyg.2021.563193. PMID: 34489769; PMCID: PMC8417359.

3) Please discuss the impact on surgeons (beside phyisicians) during the pandemic based on the following study:

Psychological Health of Surgeons in a Time of COVID-19: A Global Survey [published online ahead of print, 2021 Jan 22]. Ann Surg. 2021;10.1097/SLA.0000000000004775. doi:10.1097/SLA.0000000000004775

4) Under discussion, please mention online psychological treatment e.g. Internet Cognitive Behavior Therapy (iCBT) to help healthcare workers based on the following studies under "actionable measures should be taken into account to prevent and reduce mental health problems among healthcare worker"

The most evidence-based treatment is cognitive behaviour therapy (CBT), especially Internet CBT that can prevent the spread of infection during the pandemic.

Use of Cognitive Behavior Therapy (CBT) to treat psychiatric symptoms during COVID-19:

Mental Health Strategies to Combat the Psychological Impact of COVID-19 Beyond Paranoia and Panic. Ann Acad Med Singapore. 2020;49(3):155‐160.

Cost-effectiveness of iCBT:

Moodle: The cost effective solution for internet cognitive behavioral therapy (I-CBT) interventions. Technol Health Care. 2017;25(1):163-165. doi: 10.3233/THC-161261. PMID: 27689560.

Internet CBT can treat psychiatric symptoms such as insomnia:

Efficacy of digital cognitive behavioural therapy for insomnia: a meta-analysis of randomised controlled trials. Sleep Med. 2020 Aug 26;75:315-325. doi: 10.1016/j.sleep.2020.08.020. Epub ahead of print. PMID: 32950013.

5) The authors found that "The issue was reported by 26 respondents (11.8%) who believed that having enough

228 personal protective equipment would improve their mental health." Under discussion, please discuss the findings of the following studies and how these studies support the above observation from psychoneuroimmunological and cultural perspectives:

Is Returning to Work during the COVID-19 Pandemic Stressful? A Study on Immediate Mental Health Status and Psychoneuroimmunity Prevention Measures of Chinese Workforce [published online ahead of print, 2020 Apr 23]. Brain Behav Immun. 2020;S0889-1591(20)30603-6. doi:10.1016/j.bbi.2020.04.055

The Association Between Physical and Mental Health and Face Mask Use During the COVID-19 Pandemic: A Comparison of Two Countries With Different Views and Practices. Front Psychiatry. 2020;11:569981. Published 2020 Sep 9. doi:10.3389/fpsyt.2020.569981

**Reviewer #2: **Thank you for the opportunity to review this mixed-methods study. This study aimed to explore the incidence and influence factors of burnout syndrome, anxiety, depression, and post-traumatic disorders (PTSD) among Thai healthcare workers during COVID-19 outbreak. The following comments and suggestions may helpful for the authors to improve the quality of this manuscript. The major problem of this study is design and reliability of mixed method study.

Abstract:

(1) Please use the full names when first time mention it, such as HCWs.

(2) Methods: that would be great if the authors could add more information about the study methods, including the study setting, participants, data analysis methods and other information if needed.

(3) Results: The results must include the overall findings of the mixed-methods study rather than each sub-study.

Introduction: This part needs more work to make it clear.

(1) It would be useful to mention the background and context of the Thai healthcare system, what are the differences of the healthcare workers who working in Thailand compared with healthcare workers from other countries?

“….healthcare systems are specific to their countries”, so please add more information about this and the background of this healthcare system must be provided as many studies from other countries had been published:

Thatrimontrichai, A., Weber, D. J., & Apisarnthanarak, A. (2021). Mental health among healthcare personnel during COVID-19 in Asia: A systematic review. Journal of the Formosan Medical Association.

Marvaldi, M., Mallet, J., Dubertret, C., Moro, M. R., & Guessoum, S. B. (2021). Anxiety, depression, trauma-related, and sleep disorders among healthcare workers duirng the COVID-19 pandemic: a systematic review and meta-analysis. Neuroscience & Biobehavioral Reviews.

Li, Yufei, Nathaniel Scherer, Lambert Felix, and Hannah Kuper. "Prevalence of depression, anxiety and post-traumatic stress disorder in health care workers during the COVID-19 pandemic: A systematic review and meta-analysis." PloS one 16, no. 3 (2021): e0246454.

(2) The authors should explain more healthcare workers’ mental health during COVID-19 outbreak, as the authors mentioned that “… psychological outcomes of healthcare workers (HCWs) during the pandemic have been extensively studied globally”.

(3) Some sentences need provide reference(s), for example, “…..there was a study in Thailand finding that 42.5% of healthcare workers had anxiety”, by the way, please mention this study recruited participants during the COVID19 or not. All the information needs to focus on the topic of this study.

Methods:

(1) All the study methods related to qualitative study/methods are missing, the authors need to describe the qualitative study and the whole mixed methods design clearly.

Methodological orientation and theory are lacking. What was the reason that you chose a mixed-methods approach. It should be supported using literature. The type of mixed methods used in your study should be described.

(2) Research/study procedure is also missing. What are trained data collectors?

(3) Please provide the psychometric properties of the questionnaires used in this study.

(4) Since both qualitative and quantitative data analysis has been used, the heading should be changed to 'data analysis' instead of 'Statistical analysis'.

Results and discussion:

(1) Data integration of quantitative and qualitative results is missing. The results section is unclear if the methods of this study is unclear. It is difficult to see how it is mixed with both quantitative and qualitative data.

(2) The quotes are very brief, and it is difficult for the reader to interpret the context of these quotes.

6. PLOS authors have the option to publish the peer review history of their article (what does this mean?). If published, this will include your full peer review and any attached files.

---

## [Author Response · Author response to Decision Letter 0]

6 Mar 2022

We thank the editor and the two reviewers for their comments on our manuscript. Below is

our response to each point raised by the academic editor and reviewers. We hope that we

satisfyingly addressed them and that the manuscript will be now suited for publication.

Academic editor:

We have already ensured our manuscript meet the requirements.

2. Please provide additional details regarding participant consent

We have added details about participant consent on lines 117-118.

Reviewer #1: 

Q1: Under the Introduction, the authors stated " Since the World Health Organization declaration on March 2020 [1] the pandemic situation has been fluctuating, and the crisis has yet to be fully resolved". It is important to talk about the impact of pandemic on mental health based on the following landmark studies

We cited the data from the recommended articles in the introduction part.

1. A systematic review of COVID-19 on mental health Impact of COVID-19 pandemic on mental health in the general population: A systematic review. J Affect Disord. 2020;277:55-64. On line 56

2. The impact of COVID-19 on three continents: A chain mediation model on COVID-19 symptoms and mental health outcomes in Americans, Asians and Europeans. SciRep 11, 6481 (2021). On lines 56 and 60

3. The impact of COVID-19 pandemic on physical and mental health of Asians: A study of seven middle-income countries in Asia. PLoS One. 2021 Feb 11;16(2):e0246824. On line 56

4. Government response moderates the mental health impact of COVID-19: A systematic review and meta-analysis of depression outcomes across countries. J Affect Disord. 2021 May 27;290:364-377. On line 57

5. Association Between Mood Disorders and Risk of COVID-19 Infection, Hospitalization, and Death: A Systematic Review and Meta-analysis. JAMA Psychiatry. 2021 Jul 28. 

On lines 58 and 61

Q2: Under the paragraph, "A number of studies also identified associated factors of negative mental health outcomes 63 among HCWs". Since Thailand belongs the Asia-Pacific, please comment on the research performed in this region based on the following landmark studies but not limited to the following studies Under the discussion, the authors should compare their findings with the findings from the following Asian countries:

We cited the data from the recommended articles in the introduction and discussion part.

1. Perception Toward Exposure Risk of COVID-19 Among Health Workers in Vietnam: Status and Correlated Factors. Front Public Health. 2021 May 25;9:589317. On lines 73

2. The impact of COVID-19 pandemic on global mental health: From the general public to healthcare workers. Ann AcadMed Singap. 2021 Mar;50(3):198-199. On lines 72, 94 and 241

3. Asian-Pacific perspective on the psychological well-being of healthcare workers during the evolution of the COVID-19pandemic. BJPsych Open. 2020;6(6):e116. Published 2020 Oct 8. On lines 79, 94 and 241

4. A multinational, multicentre study on the psychological outcomes and associated physical symptoms amongst healthcare workers during COVID-19 outbreak [published online ahead of print, 2020 Apr 21]. Brain Behav Immun. 2020;S0889-1591(20)30523-7. On lines 79, 94, 239, 241, and 261-262

5. Psychological Impact of the COVID-19 Pandemic on Health Care Workers in Singapore [published online ahead of print,2020 Apr 6]. Ann Intern Med. 2020;M20-1083. On lines 79, 94 and 241

6. Impacts of COVID-19 on the Life and Work of Healthcare Workers During the Nationwide Partial Lockdown in Vietnam.Front Psychol. 2021 Aug 19;12:563193. doi: 10.3389/fpsyg.2021.563193. On lines 79

Q3: Please discuss the impact on surgeons (beside phyisicians) during the pandemic based on the following study: 

We cited the data from the recommended articles in the discussion part.

1. Psychological Health of Surgeons in a Time of COVID-19: A Global Survey [published online ahead of print, 2021 Jan22]. Ann Surg. 2021;10.1097/SLA. 0000000000004775. On lines 296-297

Q4: Under discussion, please mention online psychological treatment e.g. Internet Cognitive Behavior Therapy (iCBT) to help healthcare workers based on the following studies under "actionable measures should be taken into account to prevent and reduce mental health problems among healthcare worker"

We cited the data from the recommended articles in the discussion part on lines 285-287.

1. Use of Cognitive Behavior Therapy (CBT) to treat psychiatric symptoms during COVID-19: Mental Health Strategies to Combat the Psychological Impact of COVID-19 Beyond Paranoia and Panic. Ann Acad MedSingapore. 2020;49(3):155‐160.

2. Cost-effectiveness of iCBT: Moodle: The cost effective solution for internet cognitive

behavioral therapy (I-CBT) interventions. Technol Health Care.2017;25(1):163-165. 

3. Internet CBT can treat psychiatric symptoms such as insomnia: Efficacy of digital cognitive behavioural therapy for insomnia: a meta-analysis of randomised controlled trials. Sleep Med.2020 Aug 26;75:315-325.

Q5: The authors found that "The issue was reported by 26 respondents (11.8%) who believed that having enough 228 personal protective equipment would improve their mental health." Under discussion, please discuss the findings of the following studies and how these studies support the above observation from psychoneuroimmunological and cultural perspectives:

We cited the data from the recommended articles in the discussion part on lines 279-280.

1. Is Returning to Work during the COVID-19 Pandemic Stressful? A Study on Immediate Mental Health Status and Psychoneuroimmunity Prevention Measures of Chinese Workforce [published online ahead of print, 2020 Apr 23]. Brain Behav Immun. 2020;S0889-1591(20)30603-6. 

2. The Association Between Physical and Mental Health and Face Mask Use During the COVID-19 Pandemic: A Comparison of Two Countries With Different Views and Practices. Front Psychiatry. 2020;11:569981. Published 2020 Sep9. 

Reviewer #2: 

1. Abstract

(1) Please use the full names when first time mention it

We correct the full names following the reviewer’s comment on lines 25 and 44.

(2) Methods: that would be great if the authors could add more information about the study methods, including the study setting, participants, data analysis methods and other information if needed.

We added the recommended details of methods on lines 26-27, and 30-32.

2. Introduction

(1) It would be useful to mention the background and context of the Thai healthcare system, what are the differences ofthe healthcare workers who working in Thailand compared with healthcare workers from other countries?

We cited the data from the recommended articles in the introduction part.

1. Thatrimontrichai, A., Weber, D. J., & Apisarnthanarak, A. (2021). Mental health among healthcare personnel during COVID-19 in Asia: A systematic review. Journal of the Formosan Medical Association. On lines 68-70

2. Marvaldi, M., Mallet, J., Dubertret, C., Moro, M. R., & Guessoum, S. B. (2021). Anxiety, depression, trauma-related, and sleep disorders among healthcare workers during the COVID-19 pandemic: a systematic review and meta-analysis. Neuroscience & Biobehavioral Reviews. On lines 79

3. Li, Yufei, Nathaniel Scherer, Lambert Felix, and Hannah Kuper. "Prevalence of depression, anxiety and post-traumatic stress disorder in health care workers during the COVID-19 pandemic: A systematic review and meta-analysis." PloS one16, no. 3 (2021): e0246454. On lines 77

(2) The authors should explain more healthcare workers’ mental health during COVID-19 outbreak, as the authors mentioned that “… psychological outcomes of healthcare workers (HCWs) during the pandemic have been extensively studied globally”.

We discussed the more about this issue on lines 74-80.

(3) Some sentences need provide reference(s), for example, “…..there was a study in Thailand finding that 42.5% of healthcare workers had anxiety”, by the way, please mention this study recruited participants during the COVID19 or not. All the information needs to focus on the topic of this study.

We added detailed the more about this issue on line 96.

3. Methods

(1) All the study methods related to qualitative study/methods are missing, the authors need to describe the qualitative study and the whole mixed methods design clearly.

Methodological orientation and theory are lacking. What was the reason that you chose a mixed-methods approach. It should be supported using literature. The type of mixed methods used in your study should be described.

(2) Research/study procedure is also missing. What are trained data collectors?

(1) and (2); We added more details following the reviewer’s comment on lines 167-161.

(3) Please provide the psychometric properties of the questionnaires used in this study.

We added the psychometric properties of the questionnaires on lines 129-129, 134-134, and 139-140.

(4) Since both qualitative and quantitative data analysis has been used, the heading should be changed to 'data analysis ‘instead of 'Statistical analysis'

We changed the word “Statistical analysis” to “Data analysis” on line 149.

4. Results and Discussion

(1) Data integration of quantitative and qualitative results is missing. The results section is unclear if the methods of this study is unclear. It is difficult to see how it is mixed with both quantitative and qualitative data.

No data integration of quantitative and qualitative were executed. 

We modified the manuscript to address the points made by the editor and the reviewers. We agreed with the comments in all accounts. We believe that the manuscript is now more readable, more informative, and its conclusions more useful to the public.

---

## [Decision Letter · Decision Letter 1]

31 Mar 2022

PONE-D-21-33387R1Mental health among healthcare workers during COVID-19 pandemic in ThailandPLOS ONE

Dear Dr. Kirdchok,

Thank you for submitting your manuscript to PLOS ONE. After careful consideration, we feel that it has merit but does not fully meet PLOS ONE’s publication criteria as it currently stands. Therefore, we invite you to submit a revised version of the manuscript that addresses the points raised during the review process.

As per the reviewers' feedback, comments from reviewer 2 haven't been well addressed. In this case, further revisions are still needed before we can make a decision. Below please find the additional comments for you to address:

1. Methods: please further elaborate the rationales for choosing mixed methods this study.

2. Methods: For qualitative part, is there only one open-ended questions for qualitative data collection? any interview data? If only one open-ended question, content analysis might be quite more appropriate than the thematic analysis.

3. Methods/results: as you defined this study is a mixed methods study, but there is no any combination of the quantitative data and qualitative data. Combination is the key feature of mixed method study, otherwise it is much appropriate to define it as a multiple method study. 

We look forward to receiving your revised manuscript.

Kind regards,

Alison Wang

Academic Editor

PLOS ONE

Reviewers' comments:

Reviewer's Responses to Questions

**Comments to the Author**

1. If the authors have adequately addressed your comments raised in a previous round of review and you feel that this manuscript is now acceptable for publication, you may indicate that here to bypass the “Comments to the Author” section, enter your conflict of interest statement in the “Confidential to Editor” section, and submit your "Accept" recommendation.

Reviewer #1: All comments have been addressed

Reviewer #2: (No Response)

2. Is the manuscript technically sound, and do the data support the conclusions?

Reviewer #1: Yes

Reviewer #2: No

3. Has the statistical analysis been performed appropriately and rigorously? 

Reviewer #1: Yes

Reviewer #2: No

4. Have the authors made all data underlying the findings in their manuscript fully available?

Reviewer #1: Yes

Reviewer #2: Yes

5. Is the manuscript presented in an intelligible fashion and written in standard English?

Reviewer #1: Yes

Reviewer #2: No

6. Review Comments to the Author

Reviewer #1: Thank you for the effort and amendments.

This is an important paper during the COVID-19 pandemic. I recommend publication.

Reviewer #2: Thanks for the opportunity to review this manuscript again. I think the authors did not revise the manuscript with my comments appropriately, for example, the mixed-methods design in this study is still unclear and there are many methodological issues in this manuscript.

---

## [Author Response · Author response to Decision Letter 1]

1 Apr 2022

Academic editor:

1. Methods: please further elaborate the rationales for choosing mixed methods this study.

We choose a multiple method study design instead of mixed methods

2. Methods: For qualitative part, is there only one open-ended questions for qualitative data collection? any interview data? If only one open-ended question, content analysis might be quite more appropriate than the thematic analysis.

For the qualitative part, there are open-ended questions so, we use the content analysis instead of the thematic analysis method.

3. Methods/results: as you defined this study is a mixed methods study, but there is no any combination of the quantitative data and qualitative data. Combination is the key feature of mixed method study, otherwise it is much appropriate to define it as a multiple method study.

We define our study as a multiple method study, so there is no data integration of quantitative and qualitative were executed. In addition, we try to explain more about qualitative data in the method and discussion part. 

Reviewer #2: 

Thanks for the opportunity to review this manuscript again. I think the authors did not revise the manuscript with my comments appropriately, for example, the mixed-methods design in this study is still unclear and there are many methodological issues in this manuscript.

We define our study as a multiple method study using content analysis instead of thematic analysis because we asked only open-ended questions and there was no in-depth interview in this study. Furthermore, we try to explain more about qualitative data in the method and discussion part. 

We modified the manuscript to address the points made by the editor and the reviewers. We agreed with the comments in all accounts. We believe that the manuscript is now more readable, more informative, and its conclusions more useful to the public.

---

## [Editor Report · Decision Letter 2]

6 May 2022

Mental health among healthcare workers during COVID-19 pandemic in Thailand

PONE-D-21-33387R2

Dear Dr. Kirdchok,

We’re pleased to inform you that your manuscript has been judged scientifically suitable for publication and will be formally accepted for publication once it meets all outstanding technical requirements.

Kind regards,

Alison Wang

Academic Editor

PLOS ONE
---

## [Editor Report · Acceptance letter]

12 May 2022

PONE-D-21-33387R2 

Mental health among healthcare workers during COVID-19 pandemic in Thailand 

Dear Dr. Kirdchok:

I'm pleased to inform you that your manuscript has been deemed suitable for publication in PLOS ONE. Congratulations! Your manuscript is now with our production department. 

Kind regards, 

on behalf of

Dr. Tao (Alison) Wang 

Academic Editor

PLOS ONE